# Family Resilience and the COVID-19 Pandemic: A South African Study

Edna G. Rich *, Letitia Butler-Kruger , Inge K. Sonn , Zainab Kader and Nicolette V. Roman

The Centre for Interdisciplinary Studies of Children, Families and Society, University of the Western Cape, Robert Sobukwe Road, Bellville 7535, South Africa
* Correspondence: erich@uwc.ac.za

**Abstract:** The onset of the COVID-19 pandemic created various challenges for individuals and families across the globe. Many countries went into a state of disaster and applied strict lockdown regulations to limit the spread of the novel coronavirus. Although the sudden changes in livelihoods impacted families globally, this research is limited to understanding how families connected and resolved conflict during the pandemic. The current study therefore aimed at exploring how family dynamics and resilience in South African families were affected by the COVID-19 pandemic. This study was conducted qualitatively in the Western Cape, South Africa, with 31 participants. The results indicated that families in the Western Cape had trouble adjusting to the imposed restrictions; however, some of these families used the time they had together to adapt and find new ways of building their relationships and strengthening their bonds. The main themes indicated that the most difficult challenges were the children's schooling, financial impact from job losses, and separation from extended family members due to restrictions on movement. Furthermore, familial support and connecting as a family through open and honest communication helped the families remain resilient and fostered positive relationships.

**Keywords:** family; resilience; family relationships; coping; COVID-19; South Africa

## 1. Introduction

The novel coronavirus (COVID-19) has engulfed the lives of individuals across the globe since its initial outbreak in the Wuhan city of China in December 2019. Spreading to almost every country of the world, COVID-19 to date has affected over 300 million people and resulted in more than 5 million deaths. The rapid spread of the virus through respiratory droplets from sneezing and coughing caused the World Health Organisation (WHO) to declare the outbreak of COVID-19 a worldwide pandemic in March 2020 (World Health Organization 2020). The symptoms of the coronavirus, which are very similar to influenza, include shortness of breath, severe coughing over time, and a fever. Furthermore, global research indicates that the health systems, as well as the economic and social spheres, have been challenged by the global pandemic and continue to impact other sectors across the globe (Khetrapal and Bhatia 2020). These socio-economic factors have affected both family structure and functioning within its internal and external environments.

The effects of the pandemic on the mental health and well-being of individuals are described as profound and long-lasting (O'Connor et al. 2020). For an individual in a familial setting to meet their developmental, biological, economic, and psychological needs, their social interaction with their family is pivotal. Family relationships have always been recognised as an important aspect of the continuity of society (Günindi et al. 2012). Families are important because biological relationships and parenting are not simply about reproduction. They also serve a social function as one of the great, enduring institutions of organised human life (Günindi et al. 2012). The family systems do not simply enable individuals to grow up and find their own identity, but a family reflects the structure, culture, values, and rules of the society (Günindi et al. 2012).

The structure, culture, and values have a significant influence on the resilience of the family. In challenging times such as global pandemics, the concept of 'family resilience' is extremely relevant. Walsh (2003) refers to it as the ability to withstand and rebound from crisis and adversity, such as the COVID-19 pandemic. However, family resilience is not only about overcoming adversity; it is also about turning adversity into a catalyst for the growth of the family by supporting each other's individual needs (Walsh 2012). Thus, family resilience could help struggling individuals overcome the harsh challenges presented by the COVID-19 pandemic and in the process facilitate the growth of the family. South African families are characterised by much diversity; with many teenage pregnancies and other cultural aspects, families are often characterised by single-parenthood, child-headed households, or children being placed in the care of extended family members and grandparents (Department of Social Development [DSD] 2021). As South African families do not follow the uniform characteristics of a nuclear family (Sooryamoorthy and Makhoba 2016), the current study defines a family as two or more individuals who are related in terms of blood, adoption, marriage, or living together. In addition, South Africa is faced with greater concerns and is experiencing multiple health challenges such as larger prevalence rates of HIV/AIDS, tuberculosis, weak health systems, and increased migration. Despite the diversity of families, society needs families to raise and nurture children, keep them safe, provide for their necessities, and support their overall development, as it is said that well-functioning families promote well-developed individuals (Hall and Richter 2018).

With the start of the COVID-19 pandemic, individuals across the world had to adapt and cope with unexpected difficulties. Although a limited amount of information exists, possible challenges include constraints of movements, loss of interaction amongst members of the community and, in many instances, the loss of income. The current published information largely focuses on the nature of the disease and the potential effect on people, especially in terms of mental health, social distancing, and being quarantined. However, the pandemic influenced the way families interacted with one another, adding constraints to the interactions between individuals and community members (Nguse and Wassenaar 2021; Posel et al. 2021). Some families experienced increased conflicts, instability, and even marital breakdown resulting in divorce due to the sudden devastating pressures and negative effects that came because of the strict lockdown or mandated social distancing. Other families exploited the opportunities that this pandemic brought and established stronger ties and bonding using its positive effects (Donga et al. 2021).

In addition, adversities such as serious illnesses, unemployment, reduced income, and the uncertainty of life placed a huge amount of stress on the resilience of families and the way they cope with stress. Family resilience explains how families overcome adversity and conflict in order to do well and connect as a family (Patterson 2002; Walsh 2003). McCubbin et al. (1997) define family resilience as: " . . . the positive behavioural patterns and functional competence individuals and the family unit demonstrate under stressful or adverse circumstances, which determine the family's ability to recover by maintaining its integrity as a unit while insuring, and where necessary restoring, the well-being of family members and the family unit as a whole" (p. 5). Furthermore, Ungar (2016) identifies seven principles of family resilience, namely: (1) resilience occurs in contexts of adversity; (2) resilience is a process; (3) there are trade-offs between systems when a system experiences resilience; (4) a resilient system is open, dynamic, and complex; (5) a resilient system promotes connectivity; (6) a resilient system demonstrates experimentation and learning; and (7) a resilient system includes diversity, redundancy, and participation. These seven principles will form the methodological framework for this study. The lockdown restrictions imposed by the South African government also meant that families were now obliged to spend more time together and interact with one another. In families that were already experiencing challenges such as substance abuse and domestic violence, which is prevalent in the Western Cape province, stress levels are said to have worsened during this time. Additionally, the divorce rate in South Africa increased by 20% in 2020 (Ahmed et al. 2020). COVID-19 has not only dramatically changed the lifestyles of people but

also exacerbated the stress of working adults who have now had to simultaneously fulfil multiple roles within their families. However, little is known about the influence of a global pandemic on family relationships and interactions during the pandemic (Zeng et al. 2021; Ahmed et al. 2020; Donga et al. 2021). This study aimed to explore how the recent COVID-19 pandemic contributed to family resilience. To this end it explored the challenges families faced during the COVID-19 pandemic, giving special attention to how families connected and resolved conflict during this time of adversity.

## 2. Materials and Methods

An explorative qualitative method was utilised to explore the challenges the families in a South African sample faced during the COVID-19 pandemic. An explorative approach was chosen to gain a deeper understanding of individual's experiences of family connectedness and conflict resolution during the time of COVID-19. This study forms part of a larger study which focused on the family during the COVID-19 pandemic. Ethical approval to undertake this study was obtained by the Senate Research and Ethics Committee at the University of the Western Cape (HS20/5/4). All ethical standards, such as informed consent, voluntary participation, and confidentiality, were adhered to throughout the research process.

Snowball sampling was utilised to recruit the study's participants. Due to the restrictions imposed by the government, the information regarding the study was communicated through various websites and social media groups. Participants were recruited via social media platforms (WhatsApp and Facebook posts), websites (universities and local non-governmental organisations (NGOs)), and through a snowball effect. Telephonic interviews were considered ideal as they would limit the costs for participants and eliminate the barrier for participants who do not have access to ZOOM or Microsoft teams. For the current study, the final sample consisted of 31 adult participants who each represented a family as displayed in Table 1. Of the 31 participants, 18 were female and 13 were male. For the inclusion criteria, individuals had to be 18+ years and willing to participate in the study. The age range of the sample was 18 to 46 years and older. Many possible participants who did not have access to the websites and social media groups could have been excluded from the study. Most of the participants had a National Senior Certificate, Honours, Masters, or Doctoral degrees. Additionally, 21 were employed and 10 were unemployed. The majority of the participants lived within a nuclear family structure (two parents with one or more children).

**Table 1.** Demographic characteristics.

| Demographic Characteristics | | Frequency |
|---|---|---|
| Gender | Female | 18 |
| | Male | 13 |
| Age | 18–24 Years | 4 |
| | 25–35 Years | 18 |
| | 36–46 Years | 3 |
| | 47+ Years | 6 |
| Family characteristics | Nuclear Family | 22 |
| | Single parent family | 2 |
| | Grandparent family | 2 |
| | Extended Family | 5 |
| Educational level | High school | 10 |
| | Diploma/Certificate | 5 |
| | Bachelors/Honours | 12 |
| | Masters/Doctorate | 4 |
| Employment status | Employed | 21 |
| | Unemployed | 10 |

Semi-structured interviews were conducted with the participants. The interviews ranged from 30 to 60 min. The participants were asked about their experience of how their families connected during the pandemic and how they resolved conflict. The following questions were asked to collect information regarding the challenges and benefits experienced by families during the COVID-19 lockdown period: How would you describe your family? What do you love about your family? What do you dislike about your family? How would you describe the structure of your family (e.g., single mom, married, blended, etc.)? How do you manage conflict and relationships in your family? How does your family distribute roles? What is the biggest change you have seen in your family and in your community during this period? Do you think that the best approach has been used in South Africa to help families during COVID-19? If not, what could have been done better? How do you think families in general are coping with COVID-19? Name three positives of the COVID-19 experience as a family. Name three negatives of the COVID-19 experience as a family. Name three lessons learnt from the pandemic. How did/do the members of your family spend their time during the lockdown? What are the most important challenges your family is facing right now, and how are they handled? How do different family members handle these challenges?

Data were analysed using a thematic analysis which allows for richer and in-depth ontological and epistemological viewpoints.

Trustworthiness was achieved by ensuring external validity, reliability, and objectivity (Reason et al. 1981). A reflexive approach to the inquiry and analysis was employed to establish rigour in this study (Reason et al. 1981).

To ensure validity, the participants were informed that they could express themselves freely. Additionally, two researchers were involved in coding the information and analysing and interpreting the data in order to reach consensus. Peer debriefing and peer examination occurred at the end of each interview. To ensure reliability, the research steps were transparent and copious notes of the research process were kept; this included the methods of data collection and analysis. In addition, a single interview schedule developed by the research team was used to guide all the interviews. Objectivity was maintained by the researchers by seeking clarification from the participants when needed. This included verbatim transcripts of the participants' responses. To ensure external validity, rich and extended participant quotations were included, and a comprehensive methodology was provided.

The researchers used a reflective journal to document the discussions, deliberations, and decisions made by the research team while conducting the research. The reflective journal, which was maintained as part of an audit trail for this study, included notes involving biases, personal feelings, and insights immediately after data collection and during the process of analysis. Moreover, any experiences, preconceived perceptions, and emotions that could have affected engagement with the participants and influenced the data analysis were recorded in the reflective journal.

## 3. Results and Discussion

The onset of the COVID-19 pandemic brought unique challenges to activities of family daily living as many countries, including South Africa, imposed a hard lockdown resulting in many peoples' healthcare, education, finances, and home lives being altered in significant ways. In the current study, the results yielded the following: The participants identified changes to their children's schooling; the financial impact; and the isolation and separation of their families as the toughest challenges to have dealt with over the past few years with the implementation of the COVID-19 restrictions. These challenges placed an enormous strain on parents and caregivers, as not only did their daily routines change, but they were also expected to attend to their family's needs and, in many ways, had to fulfil multiple roles. In addition, the participants emphasised the positive impact and newfound resilience that the restrictions had on their families. Many articulated the familial support, interaction, and family resilience as positive aspects that came from the COVID-19 lockdown restrictions, as

the individuals in the family all faced the same challenges and often had to make sacrifices in order to compensate for what was now restricted.

The themes and sub-themes that emerged from the data analysis are presented in Table 2 below. Thereafter, each theme and sub-theme is discussed under its own respective sub-heading.

**Table 2.** Themes and sub-themes.

| Theme | Sub-Theme |
|---|---|
| Theme 1: Challenges experienced by the participants: | *Sub-theme 1.1:* Balancing between work and family <br> *Sub-theme 1.2:* Financial impact <br> *Sub-theme 1.3:* Separation of families |
| Theme 2: Family resilience: | *Sub-theme 2.1:* Familial support <br> *Sub-theme 2.2:* Positive familial characteristics <br> *Sub-theme 2.3:* Familial communications <br> *Sub-theme 2.4:* Conflict management |

*3.1. Theme 1: Challenges Experienced by the Participants*

Families are often faced with many challenges, but with the onset of the COVID-19 pandemic, they were confronted with a new set of difficulties, especially in homes where parents lost their income whilst trying to provide a healthy and stable environment for their children. In this regard, the participants confirmed that the pandemic brought about several challenges, such as having to balance their days between work and their families; the financial impact due to job losses and the closure of many businesses; and the separation of families due to the restriction of movement. These sub-themes are described next.

3.1.1. Sub-Theme 1.1: Balancing between Work and Family

One of the restrictions that was implemented in South Africa was the closing of schools and workplaces. Consequently, many parents had to fill the role in their children's lives as the parent, employee, and the home-schooling educator. Many individuals started working from home and attempted to fill the role as educator for their children at home. This placed enormous stress on parents as many did not have the necessary skills nor time to educate their children. Although this may have been seen by many as a burden on parents, especially mothers, it also allowed parents to spend time with their children which could have fostered the parent–child relationship and added to familial resilience (Uzun et al. 2021). The following excerpts highlight how the study's participants viewed the closure of schools and workplaces:

> *"Then you've got my son in grade five. Who's missed out, I mean they had online, they had a little bit of online during the lockdown. But to find the time to have a full-time job during the day and be a teacher at home was quite difficult."* (38, Female, Logistics Manager)

> *"Because obviously I can't divide myself, so I think that is the difficulty I struggled with, my family. It is dividing my attention and then there is not just them, it is their father as well."* (28, Female, Nail Technician and Beauty Therapist)

> *"So it was a complete shift, it was a time to find a way to be able to do own work with children at home because they are not at school and because not wanting to be willing to participate in the work, for the younger ones especially. That was a challenge, in the day in between trying to find that way of being fully working and having those demands still in place. So, it's been a big emotional roller coaster spiritually to be able to find ways to keep yourself occupied and not let it sort of in any way guide your priority because of it."* (41, Female, Private Banker)

### 3.1.2. Sub-Theme 1.2: Financial Impact

With most leisure, non-essential services, and workplaces being restricted from operating, such as bars, restaurants, holiday accommodation, hairdressers, and beauty organisations, amongst others, many individuals were left without an income and eventually lost their jobs (Luke 2020). Due to the restriction of movement and the closure of non-essential services, many companies and families were put under severe financial strain. People had to start looking for new ways to support their families and survive, as well as adapt their living expenses. This affected the current study participants in the following way:

*"I immediately closed my business doors because I was scared, and obviously that put more stress on everything else financially on top of that, so it's, it was a very, this year was just not, not a good one."* (28, Female, Nail Technician and Beauty Therapist)

*"I think in like in my own little family like me and my husband, I think probably the biggest challenge for me, for us was like financially because my husband works for himself. So, because COVID has really impacted, he was already only starting out before COVID. So already, like business was only starting to grow and with, with, with the economy the way it is, now, a lot of people were not able to afford his services. And so, he really struggled with income."* (35, Female, Attorney)

### 3.1.3. Sub-Theme 1.3: Separation of Families

Families were now restricted from moving, and this resulted in them being at home and not being able to see other members of their extended families. Many individuals rely on extended families and friends for further support and guidance. COVID-19 restrictions made it even more difficult for people to interact with others outside of the immediate family (Chersich et al. 2020). Furthermore, the restriction of interactions amongst family members were especially difficult when family members became ill and ended up in hospital or being isolated, even though many tried to keep in touch via telephone or video calls. The study's participants expressed this challenge as follows:

*"Families aren't able to see one another. When a family member is hospitalized, and even if they pass away, you're unable to be with them."* (26, Male, Unemployed)

*"The only problem is we cannot visit each other, we do not see each other, and we do miss each other, especially my sister we are very close and yes we do miss each other."* (64, Female, Pensioner)

*"You know, and when lockdown was Level 5, not being able to see anybody even though I thrived, I loved it very much. It was very difficult for some of my family."* (46, Female, Monitoring and Evaluation Officer)

The COVID-19 pandemic affected people financially, socially, and emotionally, causing further strain on individuals as it impacted everyone differently. A growing body of literature indicates a clear pattern of negative effects of COVID-19 on the mental health and well-being of individuals and families (Kelley et al. 2022). The pandemic brought unprecedented challenges for families, and they had to find new ways of adapting and coping through building family resilience.

### 3.2. Theme 2: Family Resilience

As previously mentioned, family resilience is the positive behavioural patterns and competence that the family unit demonstrate under stressful or adverse circumstances. The families in the current paper explained their family's resilience in the following ways.

### 3.2.1. Sub-Theme 2.1: Familial Support

During the pandemic, many people had to adjust to their livelihoods from what they previously knew. When caregivers are faced with highly elevated levels of stress, their mental and emotional resources are drained, making the task of positive leadership in the family challenging (Prime et al. 2020). Many families in the current study highlighted the

familial support that was given and received from one another as one factor that helped their family overcome many of the hardships brought about by the COVID-19 pandemic and this strengthened their familial resilience. This sentiment was echoed in the following excerpts:

> *"Uhm, everybody's very supportive and very understanding."* (28, Male, Software Engineer)

> *"My parents try to empower, empower us, me and my siblings in terms of education. They were so supportive. So I can say that stands out in terms of education wise."* (30, Male, PhD Graduate)

> *"Were very, very understanding of each other and very supportive of each other."* (56, Female, Tech Business Partner)

### 3.2.2. Sub-Theme 2.2: Respect between Family Members

Apart from supporting one another, they believed that having respect for one another was just as important and helped them to overcome obstacles and grow as a family. Respect is a good value to have toward one another in a family as it fosters better relationships and helps to bring positivity, peace, and calm in the surroundings, which may assist families to deal with the adversities that COVID-19 has brought about (Prime et al. 2020).

> *"Respect between the three of us."* (70, Female, Pensioner)

> *"Everyone is respectful of one another."* (35, Female, Attorney)

> *"I guess, just in general it's about trying to respect each other's space and to be mindful."* (28, Male, Software Engineer)

### 3.2.3. Sub-Theme 2.3: Positive Familial Characteristics

In the end, their family connectedness was an outcome of their ability to acknowledge one another by being supportive, loving, and respecting one another, which was strengthened by their already invested and strong family bond (Prime et al. 2020). When asked about their familial characteristics, the participants in the present study explained their family's bond in the following ways:

> *"The fact that we stick together through any trials or tribulations."* (26, Male, Unemployed)

> *"About my family, we are very close, and we get along well, and we adapt to each other's differences, and we always care about each other's opinions."* (64, Female, Pensioner)

> *"The fact that there's unity."* (35, Female, Service Representative)

### 3.2.4. Sub-Theme 2.4: Familial Communication

To understand a family, one must look at the communication that occurs between its members. In a family setting one finds affirming and incendiary communication (McCubbin et al. 1997; Prime et al. 2020). Affirming communication is positive, effective, or supportive in nature, while incendiary communication is negative or ineffective. Communication is a significant aspect of family life and may be seen as an instrument used by families to share feelings, views, needs, and preferences (Shearman and Dumlao 2008; Barnes and Olson 1985). As a contributing characteristic to family resilience, the participants mentioned that communication in their family was pivotal and applied regularly when it came to family connectedness and well-being. They confirmed this by referring to communication in the following ways:

> *"So as long as there's open communication and honesty then I don't have a—obviously if their lives are in direct danger then, ja, or if I find that it's too risky for them to do something then of course not."* (38, Female, Logistics Manager)

> *"I can say it's, it's not so hard that given that then the conflicts are resolved amicably . . . We just try to resolve it as a family. We just sit down you know as a family, everyone*

*would be there, then we discuss it well and sometimes I argue with my dad, he's a bit short-tempered but you know at the, at the same time he understands as well."* (30, Male, PhD Graduate)

*"You know, you know like shouldn't, we shouldn't, you have to rule your children, you have to have an open, yeah, it's more open, you know. It's more open."* (30, Male, Social Worker)

3.2.5. Conflict Management

As a result of the changes in the family and communication, the ways in which they previously would resolve conflict had changed and now they explained their conflict and relationship management in the following ways:

*"Well, I think with talking. You've got to talk it out obviously, uhm, I think when it comes to a marriage or even relationship with your kids you've got to also compromise. You've got to listen to both sides of the story and, ja, be realistic with your argument."* (28, Male, Software Engineer)

*"In general, I'd say we communicate quite well if there's an issue it will be raised quickly and sorted out there's, it's quite easy for us to make compromises I don't think anybody takes anything personally."* (28, Male, Software Engineer)

*"We resolve it by apologising to each other by saying or doing something wrong and we will make up by going out for the day or lunch or supper just to make up for our conflict."* (64, Female, Pensioner)

*"We manage conflict in the house by talking about the issue with each other and possibly coming up with a resolution. However, sometimes it is difficult to confront the parents about a conflict as they still hold a superior position in the family hierarchy."* (26, Female, Law Student)

Despite the challenges the families in the current study experienced, it is evident that the changes brought about by COVID-19 restrictions impacted their families both positively and negatively. Interactions and relationships among family members affect each of them. While positive relationships in a family positively affect the individuals concerned, conflicts can negatively affect social relationships and mental health. The measures taken due to the pandemic such as social distancing, isolation, and lockdown, negatively affected the social relationships of the individuals. Problems such as loneliness, anxiety, and stress may occur following a decrease in social interactions and relationships.

The current study explored how family dynamics and resilience in South African families were affected by the COVID-19 pandemic by looking at the challenges they faced during this time and how they developed as a family. The results indicated that taking on multiple roles, the financial impact, and the separation of families were some of the significant challenges that affected families the most. This indicates adversity, and according to Ungar (2016), the first principle of resilience is that it occurs in the face of adversity. These challenges cause disagreements. In this regard, the participants remarked that through familial support, respect and communication, and spending time together, they were able to overcome their differences and manage their conflict. The latter draws on the second principle which indicates that resilience is a process. The challenges experienced during the state of disaster can only be described as unique, especially in the South African context, where poverty, inequity, unemployment, and a high rate of dropouts abound.

According to UNESCO (2020), in April 2020, 188 countries implemented school closures due to the pandemic. This, amongst others, was reported as one of the major challenges the participants of the study had to face. Accompanying this challenge was the fear of the vulnerability of youth left unattended during this period. For many children with other needs, school closures meant that they did not have access to the resources offered at school, e.g., having a daily meal or other support services (Lee 2020). Many children received their only meal of the day through feeding schemes at their schools (Munje and

Jita 2019). For many children the schooling environment also offered an escape from the violence and abuse at home. Furthermore, the stress of parents and their workload also increased as they had to take on even more responsibilities. All of these changes and extra responsibilities could have potentially caused an increase in family conflict, and many had no escape from their home environments (Butler et al. 2015).

Parents and caregivers had to attempt to work remotely, or they were unable to work whilst caring for their children. This uncovers the third and fourth principles of resilience which indicate that there are trade-offs between systems when a system experiences resilience and a resilient system is open, dynamic, and complex. This added to their anxiety and stress, as the participants of this study commented that having the children at home and having to work remotely was a unique, different, and stressful challenge brought about by the pandemic. For many parents, just keeping their children busy and safe at home is a daunting prospect. For those living in low-income and divided households, this is further aggravated (Cluver et al. 2020).

Additionally, this challenge was further exacerbated by the closure of businesses and loss of jobs. The participants of the study opined that the financial strain caused by the COVID-19 pandemic was a major concern and compounded the uncertainty of having a secure income and about what the future entails. Furthermore, this economic implication of the COVID-19 pandemic resulted in impaired mental well-being (Posel et al. 2021). In addition, many families were restricted from seeing and interacting with each other. In situations where an individual was hospitalised, the rest of the family would not be able to support them physically. This has devastating effects mentally and emotionally (McDowell et al. 2020).

Furthermore, social distancing can result in increased isolation in abusive homes. In a recent systematic review by Piquero et al. (2021) it was found that incidents of domestic violence escalated in response to lockdown restrictions, a finding that was based on several studies from different cities, states, and several countries around the world. This was due to the victim being isolated from external family and friends for support and the failure of being able to escape. People in abusive homes were consequently trapped in their violent homes, and the added pressures and stress from the COVID-19 pandemic contributed to the frequency and nature of the abuse.

Despite these challenges, the participants in this study identified familial support as one of the major ways in which they resolved conflict and dealt with the challenges they encountered. COVID-19 created a severe global crisis that affected the psychological and physical health of individuals and families (Mariani et al. 2020). Families supported each other by respecting and assisting one another to be able to work or study from home. It may come across where one individual of a family experiences stress which has the potential to cross over and affect other members of the family. Parents may be at an increased risk for role strain as they adapt to changing occupation and family demands. For instance, some families may be navigating job loss and economic adversity, while others might have job security but are adjusting to new roles and expectations, such as working from home while providing childcare or home schooling, which may contribute to their familial resilience (Brock and Laifer 2020). Drawing on the principles of resilience proffered by Ungar (2018), a resilient system promotes connectivity. This was evident in the results from the participants where a resilient system demonstrated experimentation and learning, and included diversity, redundancy, and participation. For instance, when the participants of the current study worked together to overcome the diversity and hardships brought about by the COVID-19 pandemic.

The participants of this study also felt that their families were an integral part in overcoming their challenges as they empowered them to complete their education even with the closure of schools. However, although lockdown brought families closer, the participants of the current study believed that they survived these challenges because they already had a strong and invested bond as a family. These restrictions, when used

constructively, developed their family resilience and, in turn, made it easier for them to cooperate as a family.

Families assisted one another through the adversities brought about by the pandemic, and in the hard times, communication in their families was pivotal. Before the COVID-19 restrictions, families had spent more time outside of their homes and with other people. With the restrictions, they had to find ways to adapt and interact with the people in their homes. This opened the door for communication and spending quality time in getting to know each other and as seen in this study, this communication allowed for increased respect and understanding of one another. Honest communication was underscored as a necessary component in them getting through the presented adversities. Being honest about their struggles and their needs, and especially their feelings, assisted the families in further supporting each other. The COVID-19 restrictions were imposed on everyone equally, but they impacted everyone differently. Nevertheless, the open and honest communication amongst families made their family resilience against the hardships of the restrictions even stronger and more resilient.

## 4. Strengths and Limitations

One of the strengths of the current study is its qualitative inquiry in an attempt to gain an in-depth understanding of how the COVID-19 pandemic impacted family relationships and resilience. By utilising an explorative design, the researchers gained a deeper exploration of the phenomenon under study. A second strength of the study is that it was undertaken while the pandemic was being experienced and participants were presently experiencing the phenomenon.

The study, however, also had several limitations. First was the use of telephonic interviews which may have impeded the researchers' collection of data as this method limited the availability of non-verbal communication. Second was that the study participants were limited to one participant per family and therefore may be one-sided. Third, due to the interviews being conducted via telephone and Zoom, individuals who did not have access to these devices/platforms were potentially excluded. Fourth, the demographic information of the participants showed that the study's participants may have been better off than other families with limited support and access to resources; it would therefore be beneficial for future researchers to include a wider diverse population. Finally, different approaches may have been used to collect the data had there been no COVID-19 restrictions.

## 5. Implications and Recommendations

The COVID-19 restrictions continue to impact families and individuals across the globe; it is thus important that media and public organisations prioritise the effects of the pandemic on family life. Psychosocially supporting interventions focusing on families and relationships should be implemented to support families, and especially parents who assume most of the responsibility concerning the housework chores, childcare, and education in the current situation. Future guidelines on how to address stresses caused by the pandemic in the familial context should be made available to help individuals cope. This could be undertaken by raising awareness and encouraging healthy familial relationships and communication.

As a result of the pandemic, many people had to communicate, learn, and engage online. This has created an opportunity for interventions that aim to strengthen family member's resilience to occur online. This can be achieved in the form of counselling, support groups, webinars, interactive online sessions, and family conferencing. By doing this, many families can benefit, especially parents who cannot attend physical sessions due to lack of resources, or who have children at home needing attention and care. This research suggests that many families have had more opportunities to communicate and spend time together during the pandemic. It is important that once life resumes back into pre-pandemic normalcy that there is increased awareness and encouragement for families to continue to have their bonds strengthened to foster family resilience.

Important lessons have been learnt during this time and it is important for schools, government, and other institutions to take cognisance of these when they plan for the future. Hybrid models of work have proven to be effective in many institutions. This allows for family members to gain insight into the other parts of their relatives' lives. It also allows for extra time with family whilst being productive at work. Schools can utilise parents more by teaching them how to better support their children academically. This may allow the child to flourish as he/she is able to learn in a safer space with someone whom they are more comfortable with. Many people relied on hope and faith during the hardest times of the COVID-19 pandemic. Online religious services and education provided people with the opportunity to feel part of a community, to learn, and to have more family members included as they were live streamed into the home and shared with other family members. COVID-19 has provided an opportunity for resources to reach a greater audience. This practice should continue so that more people have access to resources that will foster family resilience. It would also be beneficial if researchers could follow up on the participants in order to gain insights into how the lifted restrictions influenced their families.

## 6. Conclusions

In conclusion, the participants of this study felt that despite the negative challenges that the COVID-19 pandemic brought about, the time they spent with their families allowed them to grow as individuals and as families. Additionally, it allowed family members to come together to find solutions to the problems they encountered and to support each other. Moreover, the social–emotional–psychological characteristics of the place where the family lives can directly affect all family members. While the family fulfils its own functionality, it may encounter unexpected life events and be exposed to situations beyond the family's control. As a result, some life events disrupt the balance of the family system causing an unexpected life crisis for family members. This new undesirable situation affects the functions of the family by adding different dimensions to the dynamics of the family. The study thus affirmed the positive impact COVID-19 had on family resilience and, through this research, it is hoped that more families, policy makers, and practitioners can benefit from the ideas presented herein.

**Author Contributions:** Conceptualisation, E.G.R.; Formal analysis, L.B.-K.; Methodology, L.B.-K.; Software, E.G.R.; Writing—original draft, E.G.R. and L.B.-K.; Writing—review and editing, I.K.S., Z.K. and N.V.R. All authors have read and agreed to the published version of the manuscript.

**Funding:** This research was funded by the National Research Foundation (NRF) (118581, 115460, 118551, and 129581).

**Institutional Review Board Statement:** This study was approved by the Human Social Sciences Research Ethics committee. Permission to conduct the study was granted by the Senate Research and Ethics Committee at the University of the Western Cape (Ethics Reference: HS20/5/4).

**Informed Consent Statement:** Verbal consent was obtained from the participants before the initial interview.

**Data Availability Statement:** Interview transcriptions are available at reasonable request.

**Conflicts of Interest:** The authors of this paper declare no conflict of interest.

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
