# Peer review of "Family Resilience and the COVID-19 Pandemic: A South African Study"

_socsci, doi:10.3390/socsci11090416_

Round 1
Reviewer 2 Report
Overall this was a very interesting and important study with contribution to the field. I think there are some important issues to address in the editing process:
1. The writing goes back and forth between past and present tense- particularly in the introduction which is distracting- make sure to edit the tense.
2. Should lines 93-102 be included? These look like instructions for the paper?
3. Methods- how is family defined in this study? How are connectedness and conflict resolution defined as variables and as part of family resilience. There seems to be a few different definitions of family resilience but they don't tie to the introduction and it is not clear if there were specific questions interviewers used in understanding family resilience related to these variables- please address.
4. Methods- Were there any families or participants excluded from the study? What was the criteria for exclusion?
5. The demographic data was very slim which makes it hard to understand what family's this data represents - education level, how many people in the household, who is the head of household, # of children, etc. Is there any additional data about the families you surveyed?
5. Methods- were there specific questions asked to each family in the interviews?
6. Limitations- it seems there are other limitations present in this study- for example, you only have the perspective of one member of each family. This seems like a well resourced demographic in terms of support, communication, etc. was this the case? Are there other families it would have been helpful to also hear from that were not represented in the study? Would it have been helpful to have follow up interviews if possible? Please address some of these other limitations.
Round 2
Reviewer 1 Report
Thank you for allowing me to review this article. I am impressed by the work the authors have done in amending the article in response to the reviewer's comments. The article will make a substantial contribution to the resilience literature especially within the South African context. On line 451, please change the word 'proffered' to 'offered' or 'suggested'.
Reviewer 2 Report
Overall, great improvements to the manuscript that addressed my initial concerns.